# A Case Series: Successfully Preventing COVID-19 Outbreak in a Residential Community Setting at a Drug and Alcohol Addiction Treatment Center

**DOI:** 10.3390/healthcare9010088

**Published:** 2021-01-17

**Authors:** Kenneth Hanton, Douglas McHugh, Gregory Boris

**Affiliations:** Frank H. Netter MD School of Medicine, Quinnipiac University, North Haven, CT 06473, USA; douglas.mchugh@quinnipiac.edu (D.M.); gboris@griffinhealth.org (G.B.)

**Keywords:** COVID-19, SARS-CoV-2, outbreak prevention, community, addiction, addiction rehabilitation, COVID-19 screening, screening protocol, residential

## Abstract

Coronavirus disease 2019 (COVID-19) has reduced the capacity of many addiction treatment centers, limiting access to safe, continual treatment for people with substance use disorders (SUD) in the setting of a pandemic. Here, we describe the COVID-19 screening process of a residential addiction treatment center in rural Connecticut that has had no outbreaks, closures, or reductions in capacity since the pandemic began. Out of 420 patients screened for COVID-19 from 1 February to 1 July, five patients tested positive for COVID-19: four prior to entering its residential community setting, and one after entering the residential community, resulting in no COVID-19 spread to other patients. Patient 1 presented from home and tested positive during screening prior to entry into the community. The primary care provider for patient 2 notified staff of a recent pos-itive COVID-19 test prior to the patient’s arrival on-site. Patient 3 had a COVID-19 infection in the weeks prior to arrival and tested positive during initial screening. Patient 4 tested positive af-ter coming from another addiction treatment facility that was shut down due to a COVID-19 outbreak. Patient 5 tested negative for COVID-19 during initial screening, entered the residential community, and later tested positive. It is imperative that in-person support for SUD continues during the pandemic. This case report highlights the importance of implementing a variety of tools in an effective screening process, including polymerase chain reaction screening and daily symptomology and temperature screening, which may help prevent further closures or reductions in capacity of addiction treatment centers during the COVID-19 pandemic or future outbreaks.

## 1. Introduction

The severe acute respiratory syndrome coronavirus 2 (SARS-CoV-2) virus is responsible for the coronavirus disease 2019 (COVID-19) pandemic that has spread across the globe [1]. Though vaccination efforts are currently underway, fundamental therapeutic plans and targeted treatment options for those infected with SARS-CoV-2 are limited, and management strategies at this time include mitigating the spread of the virus in the first place. This has led to many nation-wide shutdowns (although there were some exceptions) of large in-person gatherings in the United States, including many addiction-support groups. To contextualize the significance of this, it should be noted that in 2017 approximately 19.7 million American adults ages 12 and older had a substance use disorder (SUD), with 74% of cases involving alcohol, 38% of cases involving illicit drug use, and 1 in 8 adults struggling with alcohol and illicit drug use simultaneously [2]. Many with SUD are predisposed to severe COVID-19 infection due to co-morbid liver, heart, respiratory, immunological, or other organ dysfunction [3,4,5,6]. People with SUD have needed continual treatment during the COVID-19 pandemic but have faced limited access to much-needed resources because many addiction treatment centers across the U.S. were forced to reduce capacities [7,8]. Combined with increased stress, anxiety, and isolation, people with SUD are susceptible to substance misuse and relapse during the pandemic [7,9,10,11]. Therefore, it is imperative that addiction treatment, detoxification, and rehabilitation centers remain open during the pandemic [11], which is contingent upon effective screening protocols. 

Identification of a best practices model for preventing small-scale spread or outbreaks (i.e., a sudden rise in the number of cases of a disease [12]) of COVID-19 into residential communities and other care facilities is critical in achieving an ideal treatment environment, providing access to necessary resources, and preparing for subsequent waves of the virus and future pandemics [13,14]. The objective of this case series is to describe an example of a screening process at a residential addiction treatment center in rural Connecticut for patients seeking entry into its residential addiction treatment program at the beginning of the COVID-19 pandemic, which ultimately resulted in no outbreaks or small-scale spread of COVID-19 to other patients from within the treatment center. We also briefly describe five patients who tested positive for COVID-19 after presenting to the addiction recovery center at different time points in 2020 during the screening process: four of whom tested positive during screening prior to entering the residential community at the addiction treatment center and one of whom tested positive from inside the residential community after passing the screening criteria. Quinnipiac University’s Institutional Review Board judged this study to be exempt and to quality for a waiver of informed consent (protocol #06620).

## 2. Materials and Methods

The residential addiction treatment center in this case series is a 501(c)(3) not-for profit 90-bed facility located in rural Connecticut for over 80 years that offers treatment for drug and alcohol SUD and co-occurring psychiatric disorders by using a 12-step treatment regimen [15]. Each patient was assigned a therapist, a psychiatrist, and a 12-step coach. Patients enrolled in the program lived in dorms in a residential community setting within the confines of the treatment center where they were able to eat together, interact with other patients, and participate in addiction treatment programming. Prior to entering the residential community, patients were required to complete the COVID-19 screening, which took place in temporary quarantine trailers that were located within the confines of the addiction treatment center but separate from the residential community. The temporary quarantine trailers contained 2 beds, included in the 90-bed total (i.e., 2 temporary quarantine trailer beds, 88 residential community beds). Separate screening procedures for healthcare providers and staff were also implemented to prevent nosocomial transmission.

Patient data for this project were collected from the electronic medical records system in compliance with the Health Insurance Portability and Accountability Act (HIPAA) guidelines. Implementation of the COVID-19 screening process began on 1 February 2020 during the early phase of the pandemic and was modified with the addition of required mask utilization, daily temperature checks, on-site temporary quarantine trailers, nasal swab polymerase chain reaction (PCR) tests, serum antibody tests, and social distancing measures until it reached its current form that began 1 March 2020 (Figure 1).

All patients seeking entry into the addiction treatment program were required to complete the COVID-19 screening process. Between 1 February 2020 and 1 July 2020, a total of 420 new patients underwent screening. Patients came from 18 different states, with the majority from Connecticut (*n* = 277; 70.0%), New York (*n* = 82; 19.5%), and Massachusetts (*n* = 24; 5.7%). Of the 420 patients screened, 415 of them screened negative and entered the residential community. Three patients had positive COVID-19 nasal swab tests during screening prior to entry into the residential community. One patient had a negative nasal swab test during screening. One patient screened negative for COVID-19, entered the residential community, and later tested positive. The screening protocol (Figure 1) shares many components outlined by the American Society of Addiction Medicine (ASAM) COVID-19 Task Force recommendations for COVID-19 infection mitigation in residential treatment facilities [11]. Each patient must pass this series of screenings denoted in grey boxes prior to either entering the residential community (green box) or being prompted to quarantine at home before returning to the addiction treatment center (red box).

## 3. Case Presentations

Patient 1 was a man in his 50s from New York with no significant past medical history who came to the facility from home. Patient’s substance of choice was alcohol, which he began consuming casually at age 15. Patient’s alcohol consumption increased to self-medicate his depression after he and his wife became divorced. Patient had attended Alcoholics Anonymous intermittently for 2 years prior to arrival at the addiction treatment center. Patient passed the initial phone screening and temperature check but had a positive nasal swab test while in the temporary quarantine trailer. Patient was sent home to his permanent residence to quarantine for 14 days prior to returning. Patient returned post-quarantine with a negative repeat nasal swab test, was cleared by medical staff, and entered the residential community.

Patient 2 was a man in his 50s from New York with a history of chronic sinusitis, alcohol-induced blackouts, and delirium tremens who came to the facility from home. Patient’s substance of choice was alcohol, which he began consuming at age 13. Patient had been consuming 8–10 fluid ounces of rum daily with weekend binge drinking for approximately 6 years prior to arrival at the addiction treatment center. Patient passed initial phone screening and temperature check, and received a nasal swab test while in the temporary quarantine trailer. While in quarantine, prior to the return of his test results, patient’s primary care provider notified medical staff that he had a recent positive nasal swab test from home prior to arrival on site. Patient was immediately sent home to quarantine for 14 days prior to returning; however, patient failed to return to the addiction treatment center after his home quarantine. Of note, patient’s nasal swab test during screening was negative.

Patient 3 was a man in his 40s from Connecticut who came to the facility from home. Patient’s substance of choice was alcohol, with a use history consisting of two bottles of wine per day, 4–5 days per week for 2 months prior to arrival at the addiction treatment center. During initial phone screening, patient stated he had previously contracted COVID-19, which led to subsequent bilateral pneumonia, both of which cleared prior to phone screening. Patient passed initial phone screening and temperature check but had a positive nasal swab test while in the temporary quarantine trailer. Patient was sent home to his permanent residence to quarantine prior to returning. During this at-home quarantine period, patient received a series of negative repeat outpatient nasal swab test over the course of 6 days, and antibody tests performed at initial screening showed presence of serum IgG antibodies to SARS-CoV-2 virus. Patient returned to the facility 6 days after discharge, was medically cleared by staff, and entered the residential community.

Patient 4 was a woman in her 50s from New York with a history of ascites and liver cirrhosis secondary to alcoholism. Patient’s substance of choice was alcohol, which she began drinking intermittently at age 16. Patient’s alcohol use history involved consuming three or more glasses of wine daily for the past 3 years prior to arrival at the addiction treatment center. Patient sought entry after coming from a different rehabilitation facility that was shut down due to a COVID-19 outbreak. Upon arrival, patient passed initial phone screening and temperature check but had a positive nasal swab test while in the temporary quarantine trailer. Patient was sent home to her permanent residence to quarantine for 14 days prior to returning. Patient returned post-quarantine with a negative repeat nasal swab test, was cleared by medical staff, and entered the residential community. 

Patient 5 was a woman in her 60s from New York with a history of malignant lung cancer, hypertension, orthostatic hypotension, and COPD. Patient’s substance of choice was alcohol, which she began drinking heavily at the beginning of the pandemic. Patient’s use history involving consumption of two bottles of wine per day for 1 month prior to arrival at the addiction treatment center, involving consuming alcohol while working. Patient was recently at a hospital receiving cancer treatment before coming to the facility from home. Patient passed initial phone screening. Upon arrival, patient was asymptomatic with stable vital signs. Patient had a normal temperature and negative nasal swab test during quarantine in the temporary quarantine trailer for 72 h, after which she was allowed entry into the residential community. While in the residential community 16 days after arrival, patient began having symptoms of hypotension and fever. Patient was placed on supplemental O_2_ and transferred to a hospital for treatment. Hospital staff later notified medical staff at the addiction treatment center that patient was diagnosed with COVID-19. All medical staff at the addiction treatment center were subsequently informed of patient’s COVID-19 status. Patient’s roommate was immediately quarantined for 7 days in their dorm room and received a repeat nasal swab test. Repeat nasal swab test was negative, and patient’s roommate remained asymptomatic throughout quarantine. After 7 days, her roommate was medically cleared to reenter the residential community. Other residents were queried if they had been within 6 feet of the patient for more than 15 min within the past 14 days. Residents who answered ‘’yes’’ received repeat nasal swab tests the following day and were quarantined for 3 days in their dorm rooms while having their temperatures monitored. They were medically cleared to be released from quarantine and reenter the residential community after all repeat nasal swab tests were negative. All residents who were subsequently quarantined participated in programming from their dorm rooms remotely via iPads, had meals delivered to their dorm rooms, and received well-bring checks by staff. All patients at the addiction recovery center were reminded to wear masks at all times and to report any symptoms to nursing staff.

## 4. Discussion

COVID-19 has forced many in-person addiction-support resources to reduce capacity or close altogether, making it difficult for people with SUD to obtain safe, continual treatment [8]. These cases illustrate an example of a residential addiction treatment center that had no outbreaks of COVID-19 or spread of the virus to other patients within its confines, no reductions in capacity, and one confirmed case of a resident testing positive for COVID-19 from inside the residential community, which allowed it to continue to provide treatment to people with SUD during a period when relapse and overdose were increasing nation-wide [17]. 

There were many lessons learned during the development of the patient screening protocol into its current form. It was imperative that a thorough screening process was initiated from the beginning of the COVID-19 pandemic, which required early and frequent monitoring of the progression of its spread across the globe. Addiction treatment centers share with nursing homes many vulnerabilities to outbreaks, as noted by Barnett and Grabowski [18]; these include high levels of chronic illness and impairment [3,4,5,6], and people living in very close proximity. In addition to large-scale COVID-19 screening measures for patients at this study’s addiction treatment center, separate screening measures for healthcare workers and staff were also implemented to prevent nosocomial transmission. As asserted by Black et al. and Sacco et al. [19,20], screening of staff is essential for prevention of COVID-19 transmission and outbreaks among patients and protection of staff. 

During the early stages of screening protocol development, it became apparent that temperature checks and telephone pre-screening process alone were inadequate for the detection of COVID-19 among patients due to the asymptomatic nature of the virus [19,20,21]. Patients 1–5 all passed initial phone screening and temperature checks, but had a positive nasal swab PCR test, thus illustrating the necessity of early implementation of PCR testing in patient screening protocols [11,19,20,21]. Additionally, antibody testing, which was initially marketed for diagnostic purposes for COVID-19, had no value in detection of acute disease [11,16], placing a larger importance on PCR nasal swab tests and daily symptomology screenings. The ability to rapidly diagnose positive COVID-19 patients either through send-out PCR tests, or preferably on-site PCR testing, played a vital role in preventing spread. With the widespread availability of COVID-19 testing resources [22], this is a viable option for many organizations seeking to develop a robust COVID-19 screening protocol. 

Additionally, Barnett and Grabowski emphasized the importance of personal protective equipment (PPE) usage, rapid isolation of patients with suspected infection, and proactive education regarding personal hygiene in prevention of COVID-19 outbreaks in nursing homes [18]. Utilization of these strategies in the addiction recovery center, along with social distancing [11], had tremendous value in preventing viral spread in the case of Patient 5. This was demonstrated by the rapid removal and isolation of Patient 5 with the onset of symptoms, isolation and close monitoring of the patient’s close contacts, and reinforcement of the importance of wearing masks and social distancing, which likely played a role in this scenario resulting in no other instances of COVID-19 among residents.

One limitation to this study is that the nature of case reports means they provide a limited basis for generalization and replication of an identical study design is difficult. However, this case series offers insights for identifying a best practices model to prevent further COVID-19 outbreaks within other addiction treatment centers by providing a framework for COVID-19 screening based on multiple recommendations from the ASAM. These recommendations include phone screenings prior to arrival to the facility, screening new patients for COVID-19 symptomology and with RT-PCR nasal swab tests, and daily screening of current residents for fever or other symptoms indicative of a COVID-19 infection, among others discussed above [11]. Identification of an effective COVID-19 screening protocol may assist in preventing further reductions in capacity or future outbreaks in addiction treatment centers and other residential communities during the COVID-19 pandemic [13].

## Figures and Tables

**Figure 1 healthcare-09-00088-f001:**
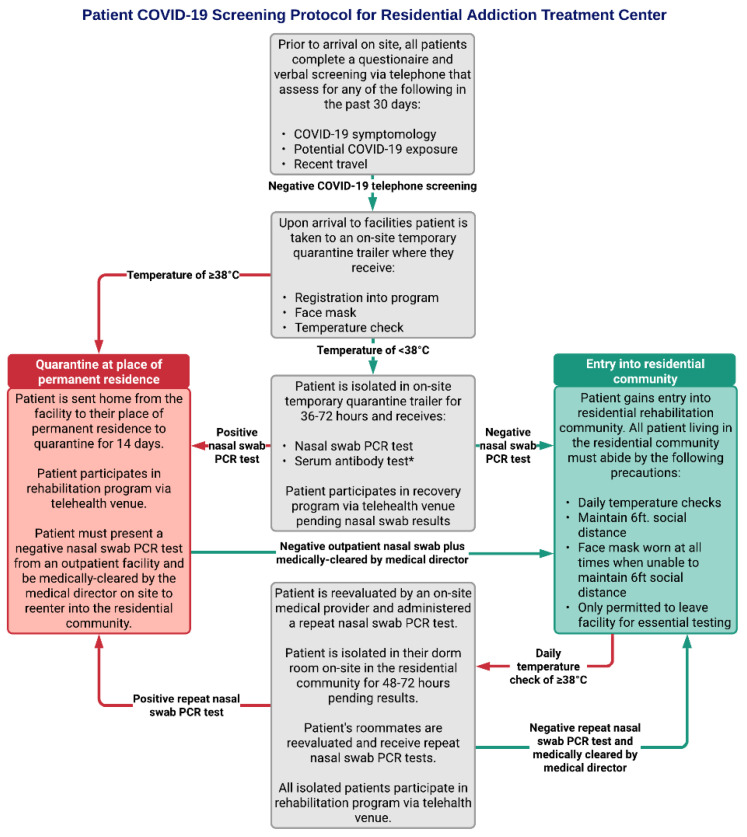
Overview of the current COVID-19 screening process for patients. * Serum SARS-CoV-2 antibody tests may have some use when informing clinical decisions, however the Centers for Disease Control and Prevention (CDC) does not recommend any change in clinical practice based on serum antibody test results as it is still unclear whether serological COVID-19 antibodies provide protective immunity against the virus [16].

## Data Availability

The data presented in this study are available on request from the corresponding author. The data are not publicly available in order to maintain participant confidentiality per IRB protocol.

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
