# Peer review of "A Case Series: Successfully Preventing COVID-19 Outbreak in a Residential Community Setting at a Drug and Alcohol Addiction Treatment Center"

_healthcare, 2021, doi:10.3390/healthcare9010088_

Round 1

Reviewer 1 Report

The topic of preventing COVID-19 in a drug and alcohol recovery residential community setting is important.   However, there is some information that I found was missing that could strengthen the paper further.  The manuscript requires some changes as indicated below. 

  1. Lines 10-11: the center is described with no outbreaks but later it is indicated that there are five positive cases. How are outbreaks defined? What is the threshold of positive cases that must be exceeded to be consider an outbreak at this facility? 
  2. Lines 28 to 29: there is a suggestion that there were “nation-wide-shutdowns of large in-person gatherings”.  I would refute this claim because in Texas, movie theaters were still open in many places since May.  Isn’t this a large gathering? Perhaps add a qualifier. E.g. “This has led to nation-wide shut-downs of large in-person gatherings, with some exceptions” etc.
  3. Lines 54 to 55: If the residents of the residential community are referred to as patients, should the ‘residential community’ be referred to as an ‘addiction recovery center’ (Line 54) or a hospital, given that there are beds? What is the difference between an addiction/recovery center/hospital and ‘residential community’.  For example, in Toronto, Canada, there is a hospital called the “centre for addiction and mental health”.  I think this distinction should be clear, perhaps expand on the definition mentioned in Line 54.  Also, why refer to it as ‘residential community’ when in fact, you first define it as an “addiction recovery center”?
  4. Line 70: the term “community” is used without “residential”. Please keep consistency.
  5. Figure 1: There are two issues. First, the title “addiction recovery residential community” sounds awkward.  Perhaps stick to “addiction recovery center”?  Secondly, is the overview of the current screening process for *patients* only? What about staff? Is there a separate screening protocol for staff? Perhaps add the words “for patients” at the end of the first sentence in Line 80 so that the distinction is clear. 
  6. Page 4: for patients 1, 2, 3, and 5, what was the addiction for each patient’s recovery/treatment? Patient 4 seemed to have alcoholism. What about the rest?  
  7. Lines 128-129: I think it should state how long patient 5 was kept in the temporary quarantine trailer e.g. 36 to 72 hours? (as mentioned in the protocol in Figure 1).    
  8. Line 140: Were other residents quarantined for 3 days in the temporary quarantine trailer? How many trailers are there? Do the trailers have beds? Are these beds included in your bed-count for the facility i.e. 90-beds in total, including trailers (line 54)?

If you decide to incorporate these revisions, please upload a manuscript that contains tracked changes or other method to highlight revisions.  Thank you for the opportunity to review this work.

Author Response

Reviewer #1

The topic of preventing COVID-19 in a drug and alcohol recovery residential community setting is important. However, there is some information that I found was missing that could strengthen the paper further. The manuscript requires some changes as indicated below.

Thank you for taking the time to review our manuscript and for providing concrete suggestions on how we can improve the information presented for future readers.

Lines 10-11: the center is described with no outbreaks but later it is indicated that there are five positive cases. How are outbreaks defined? What is the threshold of positive cases that must be exceeded to be consider an outbreak at this facility?

Thank you for the suggestion and after rereading this portion we agree that this could be made clearer. We have changed the wording in the abstract to denote that 4 of the 5 patients tested positive prior to entering the residential community with other patients, and 1 patient later tested positive from within the community after passing the initial screening, resulting in no spread of COVID-19 and no other patients becoming infected (see lines 12-14, 58-63). The Association for Professionals in Infection Control and Epidemiology defined an outbreak as, “a sudden rise in the number of cases of a disease.” We have included this definition in the manuscript for reference (see lines 50-52) and have emphasized that each patient who tested positive during the time of this study presented at different time points during screening in 2020 (see lines 58-60), meaning that the maximum number of patients with a positive COVID-19 test at the facility at any given time was 1.

Lines 28 to 29: there is a suggestion that there were “nation-wide-shutdowns of large in-person gatherings”. I would refute this claim because in Texas, movie theaters were still open in many places since May. Isn’t this a large gathering? Perhaps add a qualifier. Eg. “This has led to nation-wide shut-downs of large in-person gatherings, with some exceptions” etc.

You are correct in that there are exceptions to the shutdowns that occurred in many States across the US, and that this should be noted in our manuscript. The language has been revised to emphasize the point that there were still many large in-person gatherings that were not shut down, with in-person addiction support groups being among those that were shut down (see lines 34-36).

Lines 54 to 55: If the residents of the residential community are referred to as patients, should the ‘residential community’ be referred to as an ‘addiction recovery center’ (Line 54) or a hospital, given that there are beds? What is the difference between an addiction/recovery center/hospital and ‘residential community’. For example, in Toronto, Canada, there is a hospital called the “centre for addiction and mental health”. I think this distinction should be clear, perhaps expand on the definition mentioned in Line 54. Also, why refer to it as ‘residential community’ when in fact, you first define it as an “addiction recovery center”?

Thank you for drawing our attention to this. After rereading the mentioned sections, you’re right in that the article was lacking some description of the addiction treatment center. The addiction treatment center has been described in further detail in both the Introduction (see lines 58-63) and in the Materials & Methods section (see lines 67-78) with additional information on the layout, organization, and bed count within the facility. This should hopefully emphasize that the addiction treatment center consists of: 1) A residential community setting where enrolled patients participate in addiction treatment programming, eat, and interact with other patients 2) Separate temporary quarantine trailers where patients are screened for COVID-19 prior to entering the residential community setting

For consistency throughout the article, the facility is referred to as a “residential addiction treatment center” or “addiction treatment center” in all instances throughout the article, including the title (see lines 3-4, 8, 10, 18, 24, 43, 55-56, 62, 67, 73, 76, 103, 122-123, 133, 138-139, 144-145, 159-160, 171, 179-180, 198, 209, 212-213, 246, 253)

Line 70: the term “community” is used without “residential”. Please keep consistency.

Thank you for catching that. All instances of “community” have been changed to “residential community” for consistency, including in the title (see lines 2-3, 13-14, 20, 61-62, 71, 74, 76, 78, 95-98, 102, 127, 153, 165, 176, 184-185, 189, 201)

Figure 1: There are two issues. First, the title “addiction recovery residential community” sounds awkward. Perhaps stick to “addiction recovery center”? Secondly, is the overview of the current screening process for *patients* only? What about staff? Is there a separate screening protocol for staff? Perhaps add the words “for patients” at the end of the first sentence in Line 80 so that the distinction is clear.

Thank you for pointing this out. To maintain consistency with the wording in the rest of the article, we re-titled the figure from “COVID-19 Screening Protocol for Addiction Recovery Residential Community” to “Patient COVID-19 Screening Protocol for Residential Addiction Treatment Center”, while also denoting that this screening is for patients (see Figure 1). We also added the phrase “for patients” in the manuscript to further emphasize this point as suggested (see line 110). There was a separate screening procedure for staff members who screened positive that did not involve temporary quarantine trailers (as staff were not enrolled in the program and were not required to live on the premises for extended periods of time). While the staff screening is outside the scope of the focus of the article, we have further emphasized the point of separate screening process for patients and staff both in the introduction (see lines 78-80) and in the discussion (see lines 211-216). We hope that with these additions, the distinction is now clearer for future readers.

Page 4: for patients 1, 2, 3, and 5, what was the addiction for each patient’s recovery/treatment? Patient 4 seemed to have alcoholism. What about the rest?

This is a great point you have brought up. We have included more information pertaining to the histories of each patient, including each patient’s substance of choice and more information on their substance use histories prior to screening at the addiction treatment center (see lines 119-123, 129-133, 143-145, 157-160, 168-171).

Lines 128-129: I think it should state how long patient 5 was kept in the temporary quarantine trailer e.g. 36 to 72 hours? (as mentioned in the protocol in Figure 1).

Thank you for catching this oversight on our part. The description of patient 5 has been updated to reflect this (see lines 175-176)

Line 140: Were other residents quarantined for 3 days in the temporary quarantine trailer? How many trailers are there? Do the trailers have beds? Are these beds included in your bed-count for the facility i.e. 90-beds in total, including trailers (line 54)?

Other residents who were in contact with Patient 5 were quarantined in their dorm room (per protocol, see lines 187-188). The description of on-site temporary quarantine trailers has been included in the materials & methods section to include information about their locations and bed counts (see lines 73-78).

Reviewer 2 Report

Interesting and timely manuscript.  Well written and clear.  The manuscript can be strengthen if a few areas are re-visited and addressed.  One, the recovery residential community should be described.  For example, what kind of recovery residential community is it?  How long has it offered recovery services and what are the onsite recovery services?  Two, the case studies can bennefit from further discussion.  What kind of substances did each of the cases use, and if available, what were their drug use histories?  Three, the discussion section needs to be developed further, in paticular the second paragaph in that section.  The best practices need to identified and discussed, as found in the model.  And, how do they meet the recommendations of ASAM?   

Author Response

Reviewer #2

Interesting and timely manuscript. Well written and clear. The manuscript can be strengthened if a few areas are re-visited and addressed.

Thank you for your kind words about the paper and for taking the time to read and review it, as we always appreciate advice on how we can improve the manuscript for future readers.

One, the recovery residential community should be described. For example, what kind of recovery residential community is it? How long has it offered recovery services and what are the onsite recovery services?

Thank you for raising this point, as other reviewers similarly asked for more descriptive details. Further description of the addiction treatment center has been added to the ‘Materials & Methods’ section (see lines 67-78).

Two, the case studies can benefit from further discussion. What kind of substances did each of the cases use, and if available, what were their drug use histories?

Thank you for pointing this out, as another reviewer had the same feedback for the manuscript. Further descriptions of each patient have been included in the “Patient Presentations” section, detailing each patient’s substance of choice and substance use history prior to arrival at the addiction treatment center (see lines 119-123, 129-133, 143-145, 157-160, 168-171).

Three, the discussion section needs to be developed further, in particular the second paragraph

in that section. The best practices need to identified and discussed, as found in the model. And, how do they meet the recommendations of ASAM?

You make a great point and the other two reviewers mentioned a similar need of strengthening the discussion section with identification of both lessons learned and identification of best practices. We have added 2 additional paragraphs detailing some key lessons we learned during the development of the screening protocol, comparison to other current literature, and how those align with recommendations set forth by the American Society of Addiction Medicine (see lines 205-251). We hope that these additions, along with the citations of ASAM recommendations, make these points clearer for future readers.

Reviewer 3 Report

 A Case Series: Successfully preventing COVID-19 outbreak in a drug 2 and alcohol recovery residential community setting

Comments to the authors

Thank you for the opportunity to review this manuscript. The work presented here is important, however, there are some concerns about the manuscript in its current form that I would like to see addressed. Here are some comments that I hope will be helpful to the authors and future readers of the paper if published.

Abstract – could benefit from summaries of key lessons learnt from the screening process in the recovery residential community setting

Introduction

The introduction requires some strengthening so that it creates a great start for the reader. It should clearly make the reader know the exact objective of the paper, the identified knowledge gap that the paper addresses, and a rationale for the study. Reading through the introduction in its current state, I am unable to connect the key points communicated and some of the important aspects such as the purpose of the manuscript remain unclear to me. It also requires some editing to allow for flow of ideas.

Materials and methods

The methods section also requires some beefing up to enable the reader to understand how the study was conducted and for replicability.

Line 56 – HIPAA guidelines – abbreviation used for the first time make it easy for the reader to know exactly what the abbreviation stands for

Results

The authors give a section on case presentation describing the 5 positive cases. I believe that if the authors could add more on the results sections to clearly give some lessons learnt through the screening process that will be more useful.

Figure 1 – also requires formatting and editing

Discussion & conclusion

This section also requires some strengthening. If the authors can identify the key lessons learnt from the screening process in this facility, they will be able to have a stronger discussion of those key findings. The findings as they require some synthesis to allow for comparison with other literature. COVID-19 literature is growing, and I believe that there are other literatures that could inform the work in this manuscript.

Sacco, G., Foucault, G., Briere, O., & Annweiler, C. (2020). COVID-19 in seniors: Findings and lessons from mass screening in a nursing home. Maturitas141, 46-52.

Black, J. R., Bailey, C., Przewrocka, J., Dijkstra, K. K., & Swanton, C. (2020). COVID-19: the case for health-care worker screening to prevent hospital transmission. The Lancet395(10234), 1418-1420.

Barnett, M. L., & Grabowski, D. C. (2020, March). Nursing homes are ground zero for COVID-19 pandemic. In JAMA Health Forum (Vol. 1, No. 3, pp. e200369-e200369). American Medical Association.

Limitations

There could some study limitation that the authors should acknowledge

Author Response

Reviewer #3

Thank you for the opportunity to review this manuscript. The work presented here is important, however, there are some concerns about the manuscript in its current form that I would like to see addressed. Here are some comments that I hope will be helpful to the authors and future readers of the paper if published.

Thank you for your kind words about the importance of the work we have written about and for taking the time to review our manuscript and provide important feedback on how we can improve it.

Abstract – could benefit from summaries of key lessons learnt from the screening process in the recovery residential community setting

We have updated the abstract to reflect some key lessons learned during the development of this screening protocol (see lines 21-25), which are discussed in further detail in the discussion section (see lines 205-241)

Introduction - The introduction requires some strengthening so that it creates a great start for the reader. It should clearly make the reader know the exact objective of the paper, the identified knowledge gap that the paper addresses, and a rationale for the study. Reading through the introduction in its current state, I am unable to connect the key points communicated and some of the important aspects such as the purpose of the manuscript remain unclear to me. It also requires some editing to allow for flow of ideas.

Thank you for your feedback on how we can improve the introduction. We have added a sentence at the end of the first paragraph of the introduction (see lines 46-48), which hopefully makes for a more effective connection between the COVID-19 pandemic causing shutdowns of in-person addiction support, the necessity that in-person support groups remain open, this being contingent upon having effective screening protocols, and how our article addresses this shortcoming by describing an example of an effective screening protocol in an addiction treatment center. We have also added explicit wording in the second paragraph of the introduction identifying the objective of the manuscript (see lines 54-58).

Materials and methods - The methods section also requires some beefing up to enable the reader to understand how the study was conducted and for replicability.

We agree with your assessment here and other reviewers have made similar comments. We have included more information about the addiction recovery center program, beds, quarantine trailers, and services offered to give a better description of the addiction treatment center. We see your concern about replicability needing to be addressed and hope that our additions to the Materials & Methods has been sufficient in describing the facilities (see lines 67-80)

Line 56 – HIPAA guidelines – abbreviation used for the first time make it easy for the reader to know exactly what the abbreviation stands for

Thank you for catching this; it has been fixed (see line 83)

Results - The authors give a section on case presentation describing the 5 positive cases. I believe that if the authors could add more on the results sections to clearly give some lessons learnt through the screening process that will be more useful.

We have included more information regarding each of the 5 patients who screened positive during the screening process (see lines 119-123, 129-133, 143-145, 157-160, 168-171). Although the patient presentations for the 5 patients has been included in the study, the focus of the manuscript is on the importance of the screening protocol (Figure 1). The nature of this case series is purely descriptive, as we did not make any experimental interventions in the screening protocol, so we do not have any results or a results section per se. However, we do agree with you that the lessons learned during the development of the screening protocol should be addressed, and we have added this information to the Discussion section (see lines 205-241)

Figure 1 – also requires formatting and editing

Thank you for also pointing this out. Another reviewer suggested the title of the figure sounded awkward before and was inconsistent with the wording in the rest of the article. We have changed the title to align with the wording in the rest of the article, and noted that the screening protocol is for patients (not patients and staff). (see Figure 1)

Discussion & conclusion - This section also requires some strengthening. If the authors can identify the key lessons learnt from the screening process in this facility, they will be able to have a stronger discussion of those key findings. The findings as they require some synthesis to allow for comparison with other literature. COVID-19 literature is growing, and I believe that there are other literatures that could inform the work in this manuscript. Sacco, G., Foucault, G., Briere, O., & Annweiler, C. (2020). COVID-19 in seniors: Findings and lessons from mass screening in a nursing home. Maturitas, 141, 46-52. Black, J. R., Bailey, C., Przewrocka, J., Dijkstra, K. K., & Swanton, C. (2020). COVID-19: the case for health-care worker screening to prevent hospital transmission. The Lancet, 395(10234), 1418-1420. Barnett, M. L., & Grabowski, D. C. (2020, March). Nursing homes are ground zero for COVID-19 pandemic. In JAMA Health Forum (Vol. 1, No. 3, pp. e200369-e200369). American Medical Association.

Thank you for your comments and advice on how to strengthen the Discussion section. We have added a significant amount of information detailing the major lessons learned while developing the screening protocol. We have also read the 3 articles you suggested and have referenced each of them into the discussion section for comparison to current literature. We hope the added information strengthens the understanding of the major takeaways that were learned from developing this screening protocol, how our paper relates to other current literature about COVID-19 screening, and what specific information other institutions may be able to learn.

from this article (see lines 205-241)

Limitations - There could some study limitation that the authors should acknowledge

Thank you for prompting us to reflect further on our study’s limitations. We have included in the Discussion section that the nature of case studies means that they provide limited basis for generalization and that results may be difficult to replicate (see lines 243-244).

Round 2

Reviewer 1 Report

The topic of preventing/controlling the spread of COVID-19 in a drug and alcohol recovery residential community setting is important. The authors have revised the manuscript per the reviewer’s suggestions.  I recommend the manuscript should be accepted for publication.  Congratulations, and thank you for the opportunity to review this work.

Reviewer 3 Report

A great improvement from the first draft. Just requires minor editing but the flow of ideas in now clear in the paper